# TSPNet: Hierarchical Feature Learning via Temporal Semantic Pyramid for Sign Language Translation

**Dongxu Li**[*,1,2,3], **Chenchen Xu**[*,1,3], **Xin Yu**[1,4], **Kaihao Zhang**[1,5],
**Ben Swift**[1], **Hanna Suominen**[1,3,6], **Hongdong Li**[1,2]
The Australian National University (ANU)[1], Australian Centre for Robotic Vision (ACRV)[2],
Data61 / CSIRO[3], University of Technology Sydney (UTS)[4], Tencent AI Lab[5], University of Turku[6]
`firstname.lastname@anu.edu.au`

## Abstract

Sign language translation (SLT) aims to interpret sign video sequences into text-based natural language sentences. Sign videos consist of continuous sequences of sign gestures with no clear boundaries in between. Existing SLT models usually represent sign visual features in a frame-wise manner so as to avoid needing to explicitly segmenting the videos into isolated signs. However, these methods neglect the temporal information of signs and lead to substantial ambiguity in translation. In this paper, we explore the temporal semantic structures of sign videos to learn more discriminative features. To this end, we first present a novel sign video segment representation which takes into account multiple temporal granularities, thus alleviating the need for accurate video segmentation. Taking advantage of the proposed segment representation, we develop a novel hierarchical sign video feature learning method via a temporal semantic pyramid network, called TSPNet. Specifically, TSPNet introduces an inter-scale attention to evaluate and enhance local semantic consistency of sign segments and an intra-scale attention to resolve semantic ambiguity by using non-local video context. Experiments show that our TSPNet outperforms the state-of-the-art with significant improvements on the BLEU score (from 9.58 to 13.41) and ROUGE score (from 31.80 to 34.96) on the largest commonly-used SLT dataset. Our implementation is available at `https://github.com/verashira/TSPNet`.

## 1   Introduction

Sign language translation (SLT), as an essential sign language interpretation task, aims to provide text-based natural language translation for continuously signing videos. Since sign languages are distinct linguistic systems [1] which differ from natural languages, signed sentence and their translation into natural languages do not syntactically align. For instance, sign languages have different word ordering rules from their natural language counterparts. Because of such discrepancies between a sign language and its natural language translation, SLT methods are often required to jointly learn embedding space of sign sentence videos and natural languages as well as mappings between them, leading to a difficult sequential learning problem.

Existing SLT approaches can be categorized into *two-staged* and *bootstrapping* approaches depending on whether they require additional annotations for video and text alignments or not. Two-staged models require extra annotations, namely *gloss*, to describe sign videos with word labels in their occurring order. These models first learn to recognize gestures using gloss annotations and then rearrange the recognition results into natural language sentences. Gloss annotations significantly

---

[*]Authors contributed equally.

ease the syntactic alignment in these approaches. However, gloss annotations are not easy to acquire since they require expertise in sign languages [2]. In contrast, bootstrapping models directly learn to translate from video inputs to natural language sentences without gloss annotations. These models extend easily to a wider range of sign language resources, and have recently attracted great research interests. This paper also investigates bootstrapping methods and aims to minimize the translation accuracy gap between these two approaches by learning more expressive sign features.

Sign gestures are the minimal units that preserve semantics in sign language videos. However, because of motion blurs, fine-grained gestural details, and the transitions between different sign gestures, inferring boundaries between sign gestures is difficult. Thus, current approaches [2, 3] extract sign features in a frame-wise fashion. By doing so, only spatial appearance features are captured while neglecting the temporal dependencies between sign gestures. However, temporal information is helpful in distinguishing different signs when similar body poses appear, and therefore we expect that this information to be useful in SLT models.

In this paper, we propose a Temporal Semantic Pyramid Network (TSPNet) to learn features from video segments instead of single frames. Particularly, we aim to learn sign video representations that encode both spatial appearance and temporal dynamics. However, obtaining accurate gesture segments from a continuous sign video is difficult, while noisy segments bring substantial ambiguity for feature learning. Here, we observe two important factors impacting the semantics of sign segments. First, sign video semantics are coherent, implying that segments that are temporally close share consistent semantics locally. Second, the semantics of sign gestures are context-dependent. Namely, non-local information is helpful to disambiguate the semantics of local gestures. Motivated by these, we divide each video into segments of different granularities. Our proposed TSPNet then exploits the semantic consistency among them to further enhance sign representations. To be specific, after organizing multiple video segments of different granularities in a hierarchy, our TSPNet enforces local semantic consistency by aggregating features of segments in each *semantic neighborhood* using an inter-scale attention. When tackling local ambiguity caused by imprecise segmentation, we develop an intra-scale attention to re-weight the local gesture features along the whole video sequence. By learning features from sign segments in a hierarchical approach, our TSPNet captures temporal information in sign gestures thus producing more discriminative sign video features. As a result of stronger feature semantics, we ease the difficulty in constructing mappings between sign videos and natural language sentences, thus improving the translation results.

Our model significantly improves the translation quality on the largest public sign language translation dataset RWTH-PHOENIX-WEATHER-2014T, increasing the BLEU score from 9.58 [2] to 13.41 and ROUGE score from 31.80 [2] to 34.96, greatly relaxing the constraint on expensive gloss annotations for sign language translation models.

## 2 Related Work

**Sign Word Recognition.** Most sign language works focus on word-level sign language recognition (WSLR), aiming to recognize a single gesture from an input video [4, 5, 6, 7, 8, 9, 10, 11, 12]. Although many efforts have been devoted to WSLR, few works investigate the connections between WSLR and SLT, thus hindering the usage of WSLR models in practice. Earlier WSLR works [4, 5, 6] conduct studies on constrained subsets of vocabulary, resulting in less generalizable features. The recent research outcomes [9] show that large-scale WLSR datasets [7, 8] facilitate the generalization ability of sign feature learning. Motivated by this, we make the first attempt to apply the knowledge of WLSR models to the SLT task by reusing WLSR backbones to extract video features. Interestingly, our experiments indicate that an American Sign Language (ASL) WLSR backbone network is even effective for unseen sign languages, such as German Sign Language (GSL).

**Sign Language Translation.** A major challenge in sign language translation is the alignment between sign gestures and words of a natural language. One solution is to manually provide gloss annotations for each gesture in a video as aforementioned. However, glosses often require sign language expertise to annotate. Thus, they are expensive to label and not always available. Cihan *et al.* [2] propose a sign2text (S2T) model to predict translations directly from sign videos without glosses, which is referred to as a bootstrapping approach. In particular, their models learn video features in a frame-wise manner. Since sign gestures span over multiple continuous frames, their approach overlooks the temporal dependencies in sign gestures. Different from their work, we

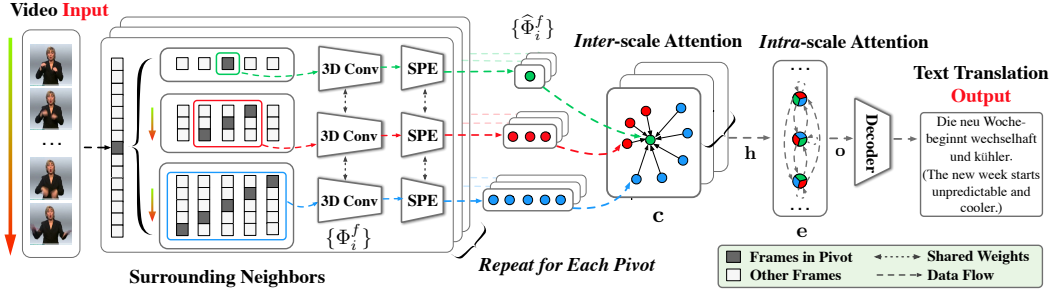

Figure 1: Overview of the workflow of our proposed TSPNet, which generates spoken language translations directly from sign language videos.

propose to learn video features from segments, and model both spatial appearance and temporal dynamics of sign gestures simultaneously. Our feature learning method exploits local and non-local temporal structure in a hierarchical way to learn discriminative sign video representations while counteracting the effect of inaccurate sign gesture segmentation.

**Neural Machine Translation (NMT).** The NMT task aims to translate one natural language to another. Most NMT models follow an encoder-decoder paradigm. Earlier works use RNN to model temporal semantics [13, 14, 15]. Later, attention mechanisms are adopted to deal with long-term dependencies [16, 17]. Instead of RNNs, the recent Transformer models [18, 19] fully rely on attention and feed-forward layers for sequence modeling. They obtain a large improvement in both translation quality and efficiency. In our work, we develop a novel encoder architecture in order to fully exploit the local and non-local semantics in sign videos, while reusing Transformer decoder to produce translations in natural language.

## 3 Temporal Semantic Pyramid Network

Our TSPNet employs an encoder-decoder architecture. The encoder learns discriminative sign video representations by exploiting the semantic hierarchical structure among video segments. The output of the encoder is fed to a Transformer decoder to acquire the translation. In this section, we first describe our proposed multi-scale segment representation for sign videos. Then we introduce the proposed hierarchical feature learning method. To focus on our main contributions, we omit the detailed decoder architecture and refer readers to [18] for reference.

### 3.1 Multi-scale Segment Representation

Previous SLT approaches [2, 3] learn video features in a frame-wise manner. Since a sign gesture usually lasts for around half a second ($\sim$12 frames) [20], these features from static images neglect the temporal semantics of gestures. Different from their approaches, we develop a segment representation for sign videos, and aim to learn both spatial and temporal semantics of sign gestures. However, as aforementioned it is difficult to obtain accurate sign gesture boundaries. To alleviate the influence of imprecise sign video segmentation, we exploit the semantic consistency among segments of different granularities in a hierarchy. Specifically, we employ a sliding window approach to create video segments with multiple window widths.

**Windowing Segments.** Given a video of $N$ frames $\mathbf{x} = \{x_0, x_1, ..., x_{N-1}\}$ with $x_i$ a video frame, a *video segment* $\mathbf{x}_{m,n}$ is a subsequence of $\mathbf{x}$, denoted as $\{x_m, x_{m+1}, ..., x_{m+n-1}\}$. For a window width $w \in \mathbb{N}$ and a stride $s \in \mathbb{N}$, we define *windowing segments* of $\mathbf{x}$ with width $w$ and stride $s$ as $\Phi(\mathbf{x}, w, s) = \{\mathbf{x}_{ks,w} \mid 0 \le k < \lfloor \frac{N}{w} \rfloor\}$.

Windowing segments evenly divide an input video into overlapping clips. However, since sign gestures in a video vary in length, it is hard to choose an appropriate window width: smaller segments

tend to capture finer-grained gestures but provide weaker contextual semantics, while larger ones are inferior to capture short gesture semantics but provide stronger context knowledge. To make the most out of the segment representation, we introduce *multi-scale segment representation* and complement the semantics of short and long segments with each other. Specifically, a multi-scale segment representation of video $\mathbf{x}$ is a set of windowing segments $\{\Phi(\mathbf{x}, w_i, s_i) \mid 0 \leq i < M\}$, where $M$, $w_i$, and $s_i$ denote the number of scales, a window width and a stride. In the following, we refer to the scales with short and long segments as small and large scales, respectively. Given the multi-scale segment representation of a video, we employ a 3D convolution network I3D [21] to extract video features for each segment. In order to adapt our backbone network to sign language gestures, we further finetune I3D on two WSLR datasets [7, 8].

## 3.2 Hierarchical Video Feature Learning for Sign Language Translation

Inaccurate sign video segmentation leads to substantial ambiguity in gesture semantics. As a result, straightforward combinations of multi-scale segments, such as pooling or concatenation, do not necessarily improve the overall translation results. To deal with the issue, we start from two key observations on the semantic structure of sign language videos, *i.e. local consistency* and *context dependency*. First, gestures in sign language videos continuously evolve. This implies that video semantics change coherently. Therefore, segments that are temporally close are expected to share consistent semantics. Second, similar sign gestures translate to different words according to the context [1, 22], and non-local video information is important for resolving semantic ambiguity in the individual gestures especially when the video segmentation is noisy. Driven by these observations, we develop a hierarchical feature learning method that utilizes local temporal structure to enforce semantic consistency and non-local video context to reduce semantic ambiguity.

Figure 1 illustrates an overview of our TSPNet. For a given sign video, we first generate its multi-scale segment representation and extract features from our I3D network $\mathcal{G}_{\mathrm{I3D}}(\cdot)$. We also develop a *Shared Positional Embedding* layer (Section 3.2.1) to inform the positions of segments in a sequence. Next, we propose to learn semantically consistent representations by aggregating features in each local neighborhood (Section 3.2.2). Lastly, TSPNet collects all the aggregated features and uses them to provide non-local video context to resolve the ambiguity of local gestures (Section 3.2.3). As an alternative, we also introduce a joint learning layer to consolidate the feature learning by utilizing local and non-local information simultaneously (Section 3.2.4). For notational convenience, we denote the windowing segments at $i$-th scale $\Phi(\mathbf{x}, w_i, s_i)$ as $\Phi_i$, and its $k$-th segment $\mathbf{x}_{ks_i, w_i}$ as $\phi_{i,k}$. We use $\phi_{i,k}^f = \mathcal{G}_{\mathrm{I3D}}(\phi_{i,k}) \in \mathbb{R}^D$ to represent the feature of the segment $\phi_{i,k}$ from the backbone, with $D$ the feature dimension.

### 3.2.1 Shared Positional Encoding

Similar to words in spoken languages, the position of a sign gesture in the whole video sequence is important for interpretation. Inspired by recent works on sequence modeling [18], we inject the position information to video segments by representing position indices in an embedding space. Specifically, we learn a function $\mathcal{G}_{\mathrm{spe}}(\cdot)$ that maps each position index into an embedding with the same length of the segment feature. Such *positional embeddings* are then added to the segment features at the corresponding position in each scale, resulting in position-informed segment representation $\widehat{\phi}_{i,k}^f = \phi_{i,k}^f + \mathcal{G}_{\mathrm{spe}}(k)$.

We pad each video by repeating the last frame to ensure the number of segments in each scale is equal. As a result, we have the same number of position indices in each scale. This allows us to share the weights of $\mathcal{G}_{\mathrm{spe}}(\cdot)$ across all the scales. By sharing weights of $\mathcal{G}_{\mathrm{spe}}(\cdot)$, we reduce the number of model parameters, especially when the number of segments is large, thus benefiting the training efficiency and alleviating overfitting when data is limited. The output of the shared positional embedding layer is the position-informed video representation in $M$ scales, *i.e.* $\{\widehat{\Phi}_0^f, \widehat{\Phi}_1^f, ..., \widehat{\Phi}_{M-1}^f\}$, where each scale has $L$ segment features $\widehat{\Phi}_i^f = \{\widehat{\phi}_{i,0}^f, \widehat{\phi}_{i,1}^f, ..., \widehat{\phi}_{i,L-1}^f\}$.

### 3.2.2 Enforcing Local Semantic Consistency

Taking advantage of the multi-scale segment representation, we tackle the issue of imperfect video segmentation by complementing smaller segments with larger but semantically relevant ones. This is

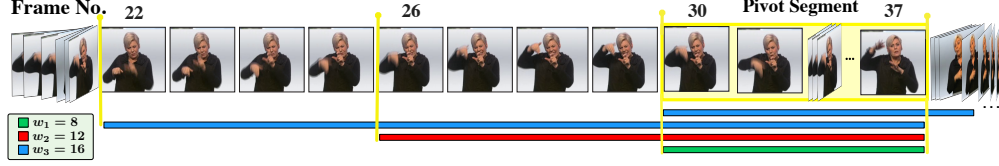

Figure 2: A surrounding neighborhood consists of features of a pivot segment and neighboring segments from multiple scales. Here we show 4 segments with the highest inter-scale attention scores. From the 26-th to 37-th frame, the GSL sign is the word **süden** (south).

achieved by an *inter-scale attention-based aggregation* of segments within *surrounding neighborhoods*. A surrounding neighborhood consists of features of one *pivot segment* (see Figure 2) from the smallest scale $\widehat{\Phi}_0^f$, and features of multiple *neighbor segments* from larger scales. Specifically, for each pivot segment, we construct its surrounding neighborhood by including segments from larger scales if their frames superset those of the pivot segment.

**Surrounding Neighborhood.** Given position-informed video representation $\{\widehat{\Phi}_0^f, \widehat{\Phi}_1^f, ..., \widehat{\Phi}_{M-1}^f\}$, with window widths arranged in an ascending order $w_0 < w_1 < ... < w_{M-1}$, we name the feature of a pivot segment, $\widehat{\phi}^f \in \widehat{\Phi}_0^f$, as a *pivot feature*. We define the *surrounding neighborhood* of a pivot feature $\widehat{\phi}^f$ as a set $\mathcal{N}_{\widehat{\phi}^f} = \widehat{\phi}^f \cup \{\widehat{\psi}^f \mid \widehat{\psi}^f \in \bigcup_{i=1}^{M-1} \widehat{\Phi}_i^f, \phi \subset \psi\}$, with $\widehat{\phi}^f, \widehat{\psi}^f$ the position-informed representation of segments $\phi$ and $\psi$, respectively.

Surrounding neighborhood imposes a containment relation between a pivot and its neighbors. As a result, we ensure gestures appearing in the pivot segment are also included in the neighbors. In this way, we use neighbors to provide more contextual clues for the pivot, and encourage the model to learn an aggregated feature that best represents the local temporal region.

**Inter-scale Attention Aggregation.** Given position-informed video features $\{\widehat{\Phi}_0^f, \widehat{\Phi}_1^f, ..., \widehat{\Phi}_{M-1}^f\}$, our hierarchical feature learning method first enforces local semantic consistency to compensate for the effect of inaccurate video segmentation. As Figure 2 shows, longer segments in the neighborhood capture more transitional gestures while shorter segments focus on fine-grained movements, both of which are helpful to recognize sign gestures in the local region. Therefore, for each pivot feature $\widehat{\Phi}_0^f$, we retrieve its surrounding neighbors and aggregate them using an *inter-scale attention*. Specifically, since segments of different scales capture different semantic aspects of the gesture, they may not reside in the same embedding space. In this regard, we first apply a linear mapping $\mathbf{W}_g \in \mathbb{R}^{D' \times D}$ to their features and map them into a shared space in $\mathbb{R}^{D'}$. We then perform scaled dot-product attention to aggregate neighbor features into the pivot feature $\widehat{\phi}_{0,k}^f \in \widehat{\Phi}_0^f$,

$$\mathcal{Z}_k = [\mathbf{W}_g z_0, \mathbf{W}_g z_1, ..., \mathbf{W}_g z_{P-1}]^T, \text{ where } z_j \in \mathcal{N}_{\widehat{\phi}_{0,k}^f}, P = \|\mathcal{N}_{\widehat{\phi}_{0,k}^f}\|, \tag{1}$$

$$c_k = \mathcal{G}_{\text{attn}}(\mathbf{W}_g \widehat{\phi}_{0,k}^f, \mathcal{Z}_k, \mathcal{Z}_k), \tag{2}$$

where $\mathcal{G}_{\text{attn}}(\cdot)$ is the scaled dot-product attention $\mathcal{G}_{\text{attn}}(\mathbf{Q}, \mathbf{K}, \mathbf{V}) = \text{softmax}(\mathbf{Q}\mathbf{K}^T/\sqrt{d})\mathbf{V}$. $\mathcal{G}_{\text{attn}}(\cdot)$ measures the correlation between $\mathbf{Q}$, $\mathbf{K}$ and uses it to re-weight $\mathbf{V}$; with $d$ the dimension of vectors in $\mathbf{K}$. The scaling factor $\sqrt{d}$ handles the effect of growing magnitude of dot-product with larger $d$ [18]. In order to learn more expressive features, we add two linear layers $\mathbf{W}_1 \in \mathbb{R}^{D \times D'}, \mathbf{W}_2 \in \mathbb{R}^{D \times D}$ with a GELU activation [23] in between as follows,

$$h_k = \mathbf{W}_2 \cdot \text{GELU}(\mathbf{W}_1 c_k + b_1) + b_2, \tag{3}$$

where $b_1$ and $b_2 \in \mathbb{R}^D$ are biases of the corresponding fully-connected layers. Each $h_k$ encodes the aggregated semantics of segments across all the scales, thus providing a locally consistent representation of the sign gestures in the temporal region.

### 3.2.3 Non-local Semantic Disambiguation

The interpretation of individual sign language gestures depends on the sentence-level context. First, the sign gesture of a word is usually a composition of two or even more "meta-signs". For example, the word "driver" requires to perform signs of "car" and "person" in order [1]. The sign of "person"

may translate to "teacher" or "student" depending on the accompanying word. Therefore, semantics for these signs are clarified only in the presence of context information. Second, there exist quite a few similar sign gestures [8]. For instance, signs of "wish" and "hungry" are very similar and are hardly distinguishable without the context. Due to the imprecise gesture segments, these ambiguities become even more severe. It is thereby important for an SLT model to consider non-local sentence information in order to resolve the semantic ambiguity. Hence, we propose to model non-local video context by an intra-scale attention over the sequence of enriched pivot features.

**Intra-scale Attention Aggregation.** After we aggregate multi-scale features into pivots, we design an intra-scale attention which takes $\mathbf{h} = [h_0, h_1, ..., h_{L-1}]^T$, $h_k \in \mathbb{R}^D$ as input, in order to enhance features across all the local regions. This is achieved by a self-attention operation on the aggregated local features, *i.e.* $\mathbf{e} = \mathcal{G}_{\mathrm{attn}}(\mathbf{W}_e\mathbf{h}, \mathbf{W}_e\mathbf{h}, \mathbf{W}_e\mathbf{h})$, where $\mathbf{W}_e \in \mathbb{R}^{D'' \times D}$ and $D''$ denotes the dimension of the hidden embedding space. Similar to the inter-scale attention, the self-attention layer is followed by two fully-connected layers for feature transformation, and then we acquire the output, namely $\mathbf{o} \in \mathbb{R}^D$. We feed the output into a Transformer decoder for translation.

### 3.2.4 Joint Learning of Local and Non-local Video Semantics

In the previous sections, the intra-scale attention follows sequentially the inter-scale attention. Therefore, the model does not have knowledge of non-local video context when enforcing local semantic consistency. This is not ideal when the non-local context is helpful for recognizing local sign gestures and easing the noisy segmentation issue. As an alternative, we propose to jointly learn local and non-local video semantics so that the two information sources interact thoroughly. In this way, non-local information helps to better recognize local gestures. In the meanwhile, enhanced local gesture semantics contribute to resolve ambiguity in turn. For this purpose, we include all the pivot segments into each surrounding neighborhood, leading to *extended surrounding neighborhood*.

**Extended Surrounding Neighborhood.** For segment representation $\{\widehat{\Phi}_0^f, \widehat{\Phi}_1^f, ..., \widehat{\Phi}_{M-1}^f\}$ of an input video, whose window widths are arranged in an ascending order $w_0 < w_1 < ... < w_{M-1}$. An *extended surrounding neighborhood* of a pivot feature $\widehat{\phi}^f$ is a set $\mathcal{N}_{\widehat{\phi}^f}^* = \mathcal{N}_{\widehat{\phi}^f} \cup \widehat{\Phi}_0^f$.

As an extended surrounding neighborhood includes semantically relevant multi-scale segments and all the pivot segments, we aggregate to learn both local and non-local sign video features, as follows,

$$\mathcal{Z}_k^* = [\mathbf{W}_c z_0^*, \mathbf{W}_c z_1^*, ..., \mathbf{W}_c z_{Q-1}^*]^T \text{ where } z_{j>0}^* \in \mathcal{N}_{\widehat{\phi}_{0,k}^f}^*, \ Q = \|\mathcal{N}_{\widehat{\phi}_{0,k}^f}^*\|, \tag{4}$$

$$c_k^* = \mathcal{G}_{\mathrm{attn}}(\mathcal{Z}_k^*, \mathcal{Z}_k^*, \mathcal{Z}_k^*). \tag{5}$$

In this way, we bring forward the non-local video context and encourage models to jointly learn to recognize sign gestures locally and mitigate the semantic ambiguity due to the inaccurate video segmentation. We eventually pass $\mathbf{c}^* = [c_0^*, c_1^*, ..., c_{L-1}^*]^T$ to two fully-connected layers to acquire the encoder output, which later passes into the Transformer decoder for generating the translation.

## 4 Experiments

### 4.1 Experiment Setup and Implementation Details

**Dataset.** We evaluate TSPNet on RWTH-PHOENIX-Weather 2014T (RPWT) dataset [2]. It is the *only* publicly available standard SLT dataset that is used for large-scale training and inference. We follow the official RPWT data partition protocol, where $7096, 519, 642$ videos are used for training, validation and testing sets, respectively. These samples are performed by nine different signers in German Sign Language (GSL) and the translations in German are also provided. RPWT dataset contains a diverse vocabulary of around 3k German words. This distinguishes SLT from most vision-and-language tasks that usually have a limited vocabulary and simple sentence structure [24, 25].

**Metrics.** We adopt BLEU [26] and ROUGE-L [27] scores, two commonly used machine translation metrics, for evaluation. BLEU-$n$ measures the precision of translation *up to* $n$-grams. For instance, BLEU-4 summarizes precision scores of 1, 2, 3 and 4-grams. We use ROUGE-L that measures the F1 score based on the longest common sub-sequences between predictions and ground-truth translations. In general, both metrics are expected to be significantly lower than 100, as there are multiple valid translations of the same meaning in natural language. This phenomenon, however, is not well quantified by existing translation metrics.

Table 1: Comparisons of translation results on RWTH-PHOENIX-Weather 2014T dataset.

| Methods | Width(s) | ROUGE-L | BLEU-1 | BLEU-2 | BLEU-3 | BLEU-4 |
|---|---|---|---|---|---|---|
| Conv2d-RNN [2] | {1} | 29.70 | 27.10 | 15.61 | 10.82 | 8.35 |
| + Luong Attn. [2]+[17] | {1} | 30.70 | 29.86 | 17.52 | 11.96 | 9.00 |
| + Bahdanau Attn. [2]+[16] | {1} | 31.80 | 32.24 | 19.03 | 12.83 | 9.58 |
| TSPNet-Single | {8} | 28.93 | 30.29 | 17.75 | 12.35 | 9.41 |
| (Transformer) | {12} | 28.10 | 29.02 | 17.03 | 12.08 | 9.39 |
| | {16} | 32.36 | 32.52 | 20.33 | 14.75 | 11.61 |
| **TSPNet-Sequential** | {8, 12, 16} | 34.77 | 35.65 | 22.80 | 16.60 | 12.97 |
| **TSPNet-Joint** | {8, 12, 16} | **34.96** | **36.10** | **23.12** | **16.88** | **13.41** |

**Implementation and Optimization.** We implement the proposed TSPNet using FAIRSEQ [28] framework in PYTORCH [29]. Since a sign gesture on average lasts around half a second ($\sim$12 frames) [20], we determine the minimal segment width to be 8 and enlarge it by $\sqrt{2}$ to another two scales, *i.e.*, 12 and 16 frame segments. In each scale, we apply a stride of 2 frames to reduce the feature sequence lengths while keeping the most semantic information. In order to extract video features, we start with a pretrained I3D networks on Kinetics [30], and then finetune it on two WSLR datasets [7, 8] in ASL to adapt to sign gesture videos. To represent texts in the feature space, we adopt SENTENCEPIECE [31] German subword embedding [32], which are based on character units to handle low-frequency words. We optimize TSPNet using Adam optimizer [33] with a cross-entropy loss as in [18, 34]. We set an initial learning rate to $10^{-4}$ and a weight decay to $10^{-4}$. We train our networks for 200 epochs, which is sufficient for all the models to converge.

## 4.2 Comparisons with the State-of-the-art

**Competing Methods.** We compare our TSPNet with two bootstrapping SLT methods. (i) *Conv2d-RNN* [2] achieves the *state-of-the-art* performance on RPWT dataset. It extracts features using AlexNet [35] and employs GRU-based [36] encoder-decoder architecture for sequence modeling. It also exploits multiple attention variants on top of recurrent units [16, 17]. We also conduct comparisons with those attention based variants. (ii) *TSPNet-Single*: in this baseline, we feed segments from only a single scale into our TSPNet. With merely single-scale feature used, it only applies intra-scale attention aggregation, which is equivalent to vanilla self-attention. As a result, this baseline degenerates to a Transformer model.

**Quantitative Comparison.** We report translation results of our TSPNet and the competing models in Table 1. The row of TSPNet-Sequential refers to the setting in Section 3.2.2 and Section 3.2.3, where we apply inter- and intra-scale attentions sequentially. TSPNet-Joint refers to the setting in Section 3.2.4, where we enhance local and non-local video semantics by jointly modeling them. As indicated in Table 1, both settings outperform the state-of-the-art SLT model, Conv2d-RNN, by a large margin, with relative improvements on BLEU-4 score by $39.80\%$ ($9.58 \rightarrow 13.41$) and on ROUGE-L by $9.94\%$ ($31.80 \rightarrow 34.96$). Benefiting from our proposed sign segment video representation, our features encode not only spatial appearance information but also the temporal information of sign gestures, and thus is more discriminative. Compared to TSPNet-Single, the multi-scale settings improve SLT performance on all metrics. This shows the effectiveness of our hierarchical feature learning. In addition, TSPNet-Joint achieves superior performance to TSPNet-Sequential. This reflects that including non-local video context is beneficial to resolve local ambiguity caused by imprecise gesture segments. Compared with previous bootstrapping approaches, TSPNet significantly narrows the performance gap between bootstrapping approaches and two-staged ones. Computationally, it costs around two hours to train a TSPNet-Joint model on a single NVIDIA V100 GPU, excluding the time for the one-off offline visual feature extraction.

**Qualitative Comparison.** Table 2 shows two example translations produced by TSPNet and the state-of-the-art model, Conv2d-RNN. In the first example, TSPNet produces a very accurate translation while Conv2d-RNN fails to interpret the original meanings. In the second example, the translation from our model retains the meaning of the sign by using the synonym of the word "rain", (*i.e.*, "shower"), while Conv2d-RNN does not capture the correct intent. However, this difference is not fully reflected on the adopted metrics. More results are provided in the supplementary material.

Table 2: Comparison of the example translation results of TSPNet and the previous state-of-the-art model. We highlight correctly translated 1-grams in blue, semantically correct translation in red.

| | |
|---|---|
| Ground Truth: | der wind weht meist schwach aus unterschiedlichen richtungen . |
| | ( mostly windy, blowing in weakly from various directions . ) |
| Conv2d-RNN [2]+[16]: | der wind weht schwach bis mäßig . |
| | ( windy, blows weak to moderate . ) |
| Ours: | der wind weht meist schwach aus unterschiedlichen richtungen . |
| | ( mostly windy, blowing in weakly from various directions . ) |
| Ground Truth: | im süden und südwesten gebietsweise regen sonst recht freundlich . |
| | ( in the south and southwest locally rain otherwise quite friendly . ) |
| Conv2d-RNN [2]+[16]: | von der südhälfte beginntes vielerorts . |
| | ( from the southpart it starts in many places . ) |
| Ours: | im süden gibt es heute nacht noch einzelne schauer . |
| | ( in the south there are still some showers tonight . ) |

Table 3: Analysis into the effects of the proposed segment representation and hierarchical feature learning method in TSPNet. We report ROUGE-L scores in R column; BLEU-$n$ in B-$n$ columns. **Left**: impact of multi-scale segments. **Right**: impact of the hierarchical feature learning method.

| $\{w\}$ | R | B1 | B2 | B3 | B4 |
|---|---|---|---|---|---|
| $\{8, 12\}$ | 30.49 | 32.04 | 19.01 | 13.42 | 10.40 |
| $\{8, 16\}$ | 33.97 | 34.58 | 21.99 | 16.10 | 12.81 |
| $\{12, 16\}$ | 33.19 | 34.03 | 21.53 | 15.66 | 12.36 |
| $\{8, 12, 16\}$ | **34.96** | **36.10** | **23.12** | **16.88** | **13.41** |

| Methods | R | B1 | B2 | B3 | B4 |
|---|---|---|---|---|---|
| Pooling | 31.21 | 32.80 | 20.17 | 14.39 | 11.13 |
| FC | 33.84 | 34.20 | 21.52 | 15.24 | 11.80 |
| NonRest. | 29.01 | 28.95 | 17.17 | 12.15 | 9.35 |
| TSPNet | **34.96** | **36.10** | **23.12** | **16.88** | **13.41** |

## 4.3 Model Analysis and Discussions

In this section, we investigate the effects of different components and design choices of TSPNet-Joint, our best model, on the translation performance.

**Multi-scale Segment Representation.** We first study the impacts of feature in different scales. As seen in Table 3, the performance of our model increases as we progressively incorporate multi-scale features. This demonstrates that learning sign representations from hierarchical features mitigates the inaccurate video segment issue. When only single-scale features are exploited, our TSPNet-Joint model degenerates to the Transformer model. We notice that the inclusion of segments with a width 16 improves the model most. This is also consistent with the finding that the 16-frame segments offer the most expressive features in a single-scale model, as indicated in Table 1. However, by incorporating segments of larger widths (*e.g.*, 24 frames), we observe a slight performance drop. This is because 24 frame segments (around 1 second) usually contain more than one sign gesture, where local semantic consistency may not hold.

**Hierarchical Feature Learning.** To investigate the effectiveness of our hierarchical feature learning method, we compare our method with three aggregation methods that do not fully consider semantic relevance among segments. (1) *position-wise pooling*: different from Section 3.2.2, we fuse features at the same temporal position across scales. Specifically, we first encode features on each scale using a separate encoder, and then apply a position-wise max-pooling operation over multi-scale segment features. (2) *position-wise FC*: we first concatenate position-wise features and then employ two fully-connected layers for aggregating features. (3) *nonrestrictive attention*: unlike Section 3.2.3, this method allows each pivot to attend to all the segments on different scales to verify the importance of enforcing local consistency. As Table 3 shows, non-structural methods fail to utilize the multi-scale segments. On the contrary, all three settings result in worse translation quality TSPNet-Single (16). This signifies the role of semantic structures when combining multi-scale segment features.

**Other Design Choices.** (i) Without finetuning the I3D networks on the WLSR datasets, the best BLEU-4 score drops to 11.23. This shows our backbone features are generalizable to even unseen sign languages (*e.g.*, GSL). (ii) When not sharing weights but learning a separate position embedding layer for each scale, we observe a slight drop of 0.08 in BLEU-4 compared to TSPNet-Joint. When we share positioning embedding layer weights, we not only reduce the model parameters but also further avoid overfitting. (iii) As indicated by TSPNet-Single (8), (12) results, simply utilizing a Transformer encoder does not achieve better performance than [2]. This implies that the performance gain mainly comes from the proposed hierarchical feature learning method. For the concern of training efficiency,

recurrent operations are rather time-consuming (up to two orders of magnitude more training time in our case). Thus, we opt to avoid using recurrent units in our model. Additionally, TSPNet-Single (16) achieves better results than [2]. This shows even with a proper uniform segmentation, our segment representation is effective in capturing temporal semantics of signs.

**Limitations and Discussions.** Although the proposed hierarchical feature learning method proves effective in modeling sign language videos, we notice several limitations of our model. For example, low-frequency words such as city names are very challenging to translate. In addition, facial expressions typically reflect the extent of signs, e.g. shower *versus* rain storm, which are not explicitly modeled in our approach. We further note that the work [37] achieves 20.17 BLEU-4 score when reusing the visual backbone networks from [38], which is reliant on the gloss annotations. In this regard, our proposed method greatly eases the requirements of expensive annotations, thus having the potential to learn sign language translation models directly from natural language sources, e.g. subtitled television footage. Finally, we remark that the proposed architecture is backbone-agnostic thus may also generalize to other tasks, such as weakly-supervised action localization [39, 40, 41]. We therefore plan to resolve the aforementioned issues and further mitigate the performance gap between gloss-free and gloss-reliant approaches as future works.

# 5 Conclusion

In this paper, we presented a Temporal Semantic Pyramid Network (TSPNet) for video sign language translation. To address the unavailability of sign gesture boundaries, TSPNet exploits semantic relevance of multi-scale video segments for learning sign video features, thus mitigating the issue of inaccurate video segmentation. In particular, TSPNet introduces a novel hierarchical feature learning procedure by taking advantage of inter- and intra-scale attention mechanism to learn features from nosily segmented video clips. As a result, the model learns more expressive sign video features. Experiments demonstrate that our method outperforms previous bootstrapping models by a large margin and significantly relaxes the requirement on expensive gloss annotations in video sign language translation.

# Acknowledgement

HL's research is funded in part by the ARC Centre of Excellence for Robotics Vision (CE140100016), ARC-Discovery (DP 190102261) and ARC-LIEF (190100080) grants. CX receives research support from ANU and Data61/CSIRO Ph.D. scholarship. We gratefully acknowledge the GPUs donated by NVIDIA Corporation, and thank all the anonymous reviewers and ACs for their constructive comments.

# Broader Impact

As of the year 2020, 466 million people worldwide, one in every ten people, has disabling hearing loss. And by the year of 2050, it is estimated that this number will grow to over 900 million [42]. Assisting deaf and hard-of-hearing people to participate fully and feel entirely included in our society is critical and can be facilitated by maximizing their ability to communicate with others in sign languages, thereby minimizing the impact of disability and disadvantage on performance. Communication difficulties experienced by deaf and hard-of-hearing people may lead to unwelcome feelings of isolation, frustration and other mental health issues. Their global cost, including the loss of productivity and deaf service support packages, is US$ 750 billion per annum in the healthcare expenditure alone [42].

The technique developed in this work contributes to the design of automated sign language interpretation systems. Successful applications of such communication technologies would facilitate access and inclusion of all community members. Our work also promotes the public awareness of people living with hearing or other disabilities, who are commonly under-representative in social activities. With more research works on automated sign language interpretation, our ultimate goal is to encourage equitable distribution of health, education, and economic resources in the society.

Failure in translation leads to potential miscommunication. However, achieving highly-accurate automated translation systems that are trustworthy even in life-critical emergency and health care situations requires further studies and regulation. In scenarios of this kind, automated sign language

translators are recommended to serve as auxiliary communication tools, rather an alternative to human interpreters. Moreover, RPWT dataset was sourced from TV weather forecasting, consequently, is biased towards this genre. Hence, its applicability to real-life use may be limited. Despite this linguistic limitation, RPWT remains the only existing large-scale dataset for sign language translation; this under-resourced area deserves more attention. Both datasets and models ought to be developed.

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
