[Reviews · NeurIPS 2020]

Review 1

Summary and Contributions: This work considers the problem of continuous sign language translation without auxiliary gloss labels. The primary contribution is a new architecture, TSPNet, which integrates cues over different temporal windows to achieve performant translation. The effectiveness of the model is evaluated on the RPWT benchmark. Update (post-rebuttal and reading the comments of the other reviewers). While there remain some concerns about the empirical evaluation of the model, and I disagree with the rebuttal claims regarding differences between sign language boundaries and action boundaries, overall I think the authors have made a useful contribution, so I maintain my marginally positive score.

Strengths: 1. Sign language translation is an important problem with the potential for significant societal impact. This work makes a meaningful advance in terms of performance on the PHOENIX benchmark. 2. The Temporal Semantic Pyramid is well-motivated as a mechanism for accounting for the nuances of sign language production. In particular, the arguments gave a compelling argument for why the model should explicitly account for both local semantic consistency and the fact that the semantics of sign gestures are context-dependent. 3. The proposed architecture is clearly communicated, through the paper (the supplementary video was also helpful here).

Weaknesses: 1. One of the motivations underpinning this work is that it is useful to avoid reliance on gloss annotations (and these are not used in training on RPWT). In practice, this should open up the ability of the model to translate any subtitled signing footage (such as the news sign footage described in [10], or other sources that could be collected). Given this ability, it would have made the story more compelling to collect a small set of such samples and show qualitative results on non-PHOENIX data. 2. The metrics suggest there is still a long way to go in terms of improving performance - it would have been useful to include an analysis of the failure cases of the model. There are a few qualitative examples in Table 2, but they are both effectively success cases. What kinds of mistakes does the proposed model make, and how might they be addressed in future work?

Correctness: The claims are principally related to the effectiveness of the proposed architecture. These are validated empirically.

Clarity: The paper is well written

Relation to Prior Work: The authors do a good job of placing the work in the context of prior efforts on sign language understanding.

Reproducibility: Yes

Additional Feedback: (To emphasize, I assigned this comment zero weight in my review, since the work in question appeared on arXiv only slightly more than 2 months before the submission deadline, and according the NeurIPS reviewer guidelines: "Note that authors are excused for not knowing about all non-refereed work (e.g, those appearing on ArXiv). Papers (whether refereed or not) appearing less than two months before the submission deadline are considered contemporaneous to NeurIPS submissions; authors are not obligated to make detailed comparisons to such papers (though, especially for the camera ready versions of accepted papers, authors are encouraged to).") But as a suggestion (for a camera ready version), it might be useful to include a comparison to the CVPR20 sign language transformers work of https://arxiv.org/abs/2003.13830, and to offer a perspective on the relative difference in performance (they achieve 20.17 BLEU4) between the two approaches.


Review 2

Summary and Contributions: The paper proposes a video model for sign language translation. Because different signs can span a variable number of frames and sign boundaries are generally not available as labels to first train a segmentation model, the authors propose to use overlapping sub-clips at multiple temporal scales (together with position embeddings). I3D is used as feature encoder for the sub-clips. To combine the information from the different clips at different scales, the authors use self-attention within each scale and across scales. Results on a large-scale SLT benchmark show improvements over variants of conv2d+RNN baselines.

Strengths: The problem of chunking videos into semantically meaningful parts without explicit supervision is a very challenging and impactful one in general. And making sense of long-range dependencies to integrate coherently the information from different chunks is also important. The proposed model based on multiple temporal scales and self-attention is sensible. The results are improved compared to baselines.

Weaknesses: I would have liked to see more discussion comparing the proposed model with other video models that try to integrate multi-scale features, e.g. SlowFast network for action recognition. L326: "simply utilising a Transformer does not achieve better performance than baseline, hence the perf gain comes from the hierarchical processing of features." That is true for 8 and 12 frames, but the 16 frames Transformer is already better than the baseline for all scores (Table1). And actually the next statement highlights that Transformer with 16 is better than baseline, hence better to not use RNNs. But then the 2 claims are somewhat contradictory. So I would recommend that you modify the first claim. Maybe one missing experiment in the ablation is to have a model using a 3d encoder but without attention (e.g. SlowFast mentioned above), to understand if you gain something by just using a 3d encoder as opposed to the baseline that uses a 2d encoder (followed by rnn). At L328, the authors mention that at training time, the 3D model is faster than the RNN based model. However, 2dconvs+RNNs might be more amenable to online causal operation mode during inference compared to the non-causal 3D convolutions. This might be relevant for SLT, so would be good to include a discussion on this. AFTER REBUTTAL: I have read the authors' rebuttal and the other reviews. I maintain my original score, although I would not mind if the paper does not get accepted. To validate fully the proposed design, a thorough comparison should be done with video models for action recognition or action detection. I do not agree that the boundaries for actions are clear, as the authors stated in the rebuttal. Even if annotations exist for action boundaries for some datasets, the proposed method could be run ignoring those labels and see if the proposed design generalises to other datasets/tasks.

Correctness: I think so.

Clarity: Yes.

Relation to Prior Work: As mentioned in Weaknesses, I would have liked to see throughout the paper more discussion comparing the proposed work with more generic video models, e.g. SlowFast which has the notion of temporal scale explicitly through the 2-stream design of the network. Or Temporal Segment Network. The current Related Work section covers only sign language translation works. The proposed solution would gain by presenting it in a broader context as well.

Reproducibility: Yes

Additional Feedback: L330 Typo: "Thus we opt to not avoid using recurrent units." there is an extra word there.


Review 3

Summary and Contributions: The work proposed introducing temporal information to encode sign language, an an improvement that rely on frame-based approaches. This is achieved by dividing the videos in segments of different temporal granularities. The multiscale representations of each pivotal frame are later combined with attention mechanisms that all allow a contextual interpretation of the different signs. The model is engineered to facilitate the recognition of individual signs in the context of continual sign language by proposing temporal windows of approximately the duration of a sign. The reported results improve the state of the art in the RWTH-PHOENIX-Weather 2014T dataset, which is currently the most important for the sign language translation task.

Strengths: S1 The work obtains a state of the art performance in direct sign to text translation in the RWTH-PHOENIX-Weather 2014T dataset, which is the most popular at the moment for the task of sign language translation. S2 The work proposes a video encoder architecture that has been engineered taking into account the durations and context of sign language, so the knowledge domain has been applied when designing the architecture. This way, the model has the potential to approximately adjust to the sign boundaries even when no supervised annotation at training time is provided. S3 The proposed architectures allows an end-to-end training for both the spatial and temporal dimensions. S4 Ablation studies support the gains achieved in the different contributions.

Weaknesses: W1 The submission claims that existing approaches only capture spatial appearance (line 42), but the one that is compared with [2] is actually based on RNNs, that have the potential to capture motion information across a sequence of frames. W2 While the work acknowledges the challenges of of motion blurs and fine-grained gesture details (line 40), it does not address them in the proposed approach. W3 The quantitative gains in terms of BLEU (9.58 to 13.41) and ROUGE (31.80 to 34.96) scores are not outstanding. W4 The results of [2] by exploiting the glosses available in the dataset are better than the ones in this submission. Given that the contributions of the work address the visual representation, it is not argues why the proposed techniques are also assess with the Sign-to-Gloss-to-Text set up considered in [2]. W5 The proposed technique would have been better assessed in an action detection/localization benchmark. Combining the main contribution in video encoding with a very specific task of sign language translation does not seem the best benchmark to evaluate the presented contributions. The related work should have been addressed comparing different video encoding schemes for action detection/localization in this set up of sign language translation. W6 While the submission includes an ablation study in which no hierarchical features are considered (TSPNet-Single) and its performance is worse than including the hierarchical architecture, a discussion about the additional computational requirements and model capacity is also missing. W7 The work actually uses a transformer as a decoder, which is not a fair comparison either with [2], which used RNN+attention. A fair comparison would have been including the proposed encoder in the RNN+attention set up. Otherwise, the gains may be mainly thanks to the transformer decoder instead of the proposed encoding scheme. W8 The work compared with [2] used as a visual encoder GoogleNet, which was actually adopted but [2]'s authors in their previous work: Koller, O., Zargaran, S., & Ney, H. (2017). Re-sign: Re-aligned end-to-end sequence modelling with deep recurrent CNN-HMMs. In Proceedings of the IEEE Conference on Computer Vision and Pattern Recognition (pp. 4297-4305). Actually, the submission states that the visual features were extracted with AlexNet (line 269), but the work from Koller et al 2017 writes "After comparing different CNN architectures [34, 24, 37], we opted for the 22 layer deep GoogleNet [37] architecture, which we initially pre-train on the 1.4M images from the ILSVRC-2012" In the submission they are using I3D [19] at different temporal proposals (window segments), pre-trained on two WSLR datasets [8,9]. **** AFTER READING THE REBUTTAL *** IN-THE-WILD EXPERIMENTS (related to W4): I was not asking for collecting more glosses as argued in the rebuttal, but using the glosses already available in the RTWH Weather dataset to show that the proposed encoding scheme also brings gains when glosses are available. So, I was asking for exactly the same experiments as in [2] with whom you compare with. If the hypothesis are correct, gains should also be observed there. ALTERNATIVE LANGUAGE DECODER (related to W7): I appreciate the effort of obtaining the ablation study of replacing the transformer decoder with an RNN. The reported results in the rebuttal support the main claims of the paper. It is actually surprising that results even improve a bit the BLEU-4 score, as I would have expected that the transformer architecture would always outperform the RNN+attention one. This is an interesting finding. CLARIFICATION OF BACKBONES (related to W8): Thank you for the clarification. My point on the weakness is that, independently from the multi-scale and attention mechanisms which built the core of the proposed contributions, the work is actually comparing a powerful I3D encoder pre-trained with two sign language datasets with [2], which uses very simple AlexNet features. So, similarly to W7, a fair comparison would have been implementing the proposed contributions over AlexNet-like features (I am aware that I3D is already considering 3D convolutions that Alexnet does not have, but you may inflate the AlexNet features and develop the multi-scale attention mechanism on them). I understand that you present your gains on top of the TSPNet-Single as a baseline, which contains the pre-trained I3D encoder. I am actually surprised that this encoder is not bringing larger gains to any AlexNet-based feature extractor + RNN, but I understand this is not the central discussion in the submission. USER STUDIES (related to W3): Thank you for completing the automatic metric with a user study. The reported results seem convincing and are actually more valuable that the BLEU and ROUGE scores as they have been assessed by humans. DIFFERENCE BETWEEN SLT AND GENERIC ACTION LOCALIZATION (related to W5): Thank you for the discussion about how the action detection task differs with sign language recognition. While I am not fully convinced that "temporal boundaries of an activity are evidently clear" in all cases, I can understand your claim. Introducing the idea of working with weak supervised action localization is also appreciated as an alternative that may benefit from the proposed architecture. COMPUTATION COST (related to W6): Thank you for reporting that the architecture take 110 ms on average to process a video on a NVIDIA V100 GPU. (Extra, not considered in the review score or decision:) DISCUSSION WITH CIHAN 2020: The Sign2Text set up considered in this work does not considers glosses, so it would be comparable with the results of this submission. So the question of why the obtained results are below CIHAN 2020 is still open, even if ignored for this review, discussion and decision.

Correctness: The claims are correct in the sense that the proposed I3D video encoder applied at different temporal scales combined with a transformer decoder is better than the GoogleNet image encoder pretrained on ImageNet combined with an RNN+attention decoder. The gains are reflected in the reported results, but it is difficult to assess where the gains come from.

Clarity: Yes, the authors have provided context and clear explanations for the presented contributions.

Relation to Prior Work: The main weakness of this submission is that it does not consider the current state of the art for sign language translation in the same dataset that it is reported: Camgoz, N. C., Koller, O., Hadfield, S., & Bowden, R. (2020). Sign Language Transformers: Joint End-to-end Sign Language Recognition and Translation. In Proceedings of the IEEE/CVF Conference on Computer Vision and Pattern Recognition (pp. 10023-10033). This work was released on arXiv on March 30, 2020 (https://arxiv.org/abs/2003.13830) and CVPR was hold on later June (after NeurIPS deadline for submission. Following the NeurIPS reviewer regulations, not including this work in the submission is accepted, so this is not an argument for penalization or rejection. In any case, if being accepted, it would be advisable to discuss how the proposed approach differs from this state of the art, especially with a quantitative results much lower than this state of the art. Actually, the submission basically compares with the work from the same authors from 2 years before, referred as [2] in the text: Cihan Camgoz, N., Hadfield, S., Koller, O., Ney, H., & Bowden, R. (2018). Neural sign language translation. In Proceedings of the IEEE Conference on Computer Vision and Pattern Recognition (pp. 7784-7793). Regarding the division of video in segments of different lengths, the authors should consider citing previous work working with action proposals for action detection/localization, which is a task that seems related to the proposed approach: Shou, Zheng, Dongang Wang, and Shih-Fu Chang. "Temporal action localization in untrimmed videos via multi-stage cnns." CVPR 2016. Liu, Y., Ma, L., Zhang, Y., Liu, W., & Chang, S. F. (2019). Multi-granularity generator for temporal action proposal. In Proceedings of the IEEE Conference on Computer Vision and Pattern Recognition (pp. 3604-3613).

Reproducibility: Yes

Additional Feedback: I am especially interested in your results with the TSPNet-Single (Transformer) encoder, which are way below that the ones recently reported in the CVPR 2020 paper: Camgoz, N. C., Koller, O., Hadfield, S., & Bowden, R. (2020). Sign Language Transformers: Joint End-to-end Sign Language Recognition and Translation. In Proceedings of the IEEE/CVF Conference on Computer Vision and Pattern Recognition (pp. 10023-10033). ...even when they are using C2D frame features extracted with GoogleNet instead of the supposed-to-be more powerful I3D features you consider. While I agree that you should not be penalized for not including this CVPR 2020 in your submission, could you please clarify what is the different between the two set ups and why your results are so much worse than theirs ?


Review 4

Summary and Contributions: The authors proposed a sign language translation method, called TSPNet. The proposed method represents video segments using multi-scale sliding window to make use of the temporal information of the video. Both inter-scale and intra-scale attention are jointly learned from the video segments to improve the feature discrimination. The experimental results validate that the proposed method improves the translation accuracy.

Strengths: The temporal information of video is captured in multi-scales. The inter-scale attention and intra-scale attention are considered simultaneously. The experimental accuracy is good.

Weaknesses: The authors proposed the problem that previous frame-wise methods cannot segment the videos into isolated signs, and propose to explore the temporal semantic structures of sign videos. However, the proposed method only makes use of multi-scale temporal windows, which is another kind of sliding window. Its temporal consistency is a regular mode as well. Although this method samples video clips and extracts features in multiple granularities, the video cannot be segmented into isolated signs and construct temporal semantic structure. In the experiments, the proposed method is evaluated to improve the accuracy of translation from the results, but it cannot be proved that the isolated signs can be segmented clearly. The construction of semantic structure is not validated as well.

Correctness: The claims and method are correct. The empirical methodology is correct.

Clarity: Yes. The paper is well written.

Relation to Prior Work: It is clearly discussed how this work differs from previous contributions in the section of Introduction and Related Work.

Reproducibility: Yes

Additional Feedback:

[Author Response · NeurIPS 2020]

We thank all the reviewers for their constructive comments. Below, we address concerns raised by reviewers.

**In-the-wild experiments (R1, R3)**. Because of our insufficient domain knowledge on German signs languages, we opt to collect 76 short videos in German sign language from the internet as in [10]. In this way, we ensure our model trained on RPWT fully covers the vocabularies, thus facilitating the validation of translations. Results show that the proposed TSPNet correctly translates 32 phrases despite the domain gap. Although there are a large number of sign videos, annotating them with glosses requires expert knowledge while being time-consuming and laborious. Without such gloss annotations, our method is still able to construct a sign language translation model. Moreover, the data for training our model are easy to collect similar to [10]. These reasons make our method more favorable.

**Discussions on Cihan et al., 2020 (R1, R3)**. Cihan et al. employ a visual encoder finetuned on the RWTH-PHOENIX dataset (GSL) *with gloss annotations*. Training/finetuning on gloss annotations will facilitate a model to effectively infer sign boundaries. In other words, Cihan et al.'s method requires gloss annotations of a particular dataset to achieve its best performance. Thus, comparing our results with theirs is unfair since we do not use gloss annotations.

**Current limitations (R1)**. (i) We notice that low-frequency words, such as city names, are very challenging to translate, we plan to provide our models with an external knowledge database and allow it to search similar gestures for translation. (ii) Facial expressions often reflect the extent of an event, e.g. shower *vs.* rain storm, which are currently not well interpreted by our models. Our future work will try to incorporate facial landmarks to improve translation accuracy. We will include the above discussions in the revision.

**Alternative visual backbone and language decoder (R2, R3)**. (i) TSPNet aims at proposing a generic learning framework to translate continuous signing videos to natural language sentences directly. We adopt I3D as visual feature backbone considering its recent success in sign language interpretation [8, 9, 10]. As SlowFast network demonstrates its superior action modeling capacity to I3D, we believe employing a stronger visual backbone, like SlowFast network, would lead to better translation results. Due to the tight timeframe of rebuttal, we cannot report results using the SlowFast network. We will add the results and discuss the impact of alternative visual encoders in the revised version. (ii) As suggested by R3, after replacing a Transformer decoder with an RNN in [2], we did not observe performance changes in BLEU-4 (+0.09) or ROUGE-L (-0.03). This implies that the performance gains mainly come from the proposed hierarchical encoding scheme rather the decoder.

**Clarification on backbones** (i) (**R2, R4**) We replace Conv2d-based features with I3D features into an RNN model and obtain BLEU-4 of 11.27 and ROUGE-L of 32.22, similar to TSPNet-Single. Since our I3D backbone captures the spatial-temporal clues from video segments, it will leverage neighboring sharp frames to infer motion blurred sign gestures. In addition, our hierarchical feature learning effectively exploits windowing segments at different granularities, facilitating to capture fine-grained gestures. (ii) (**R3**) We clarify [2] uses AlexNet (see Sec. 5 in [2]), while Koller et al. and Cihan et al. use GoogleNet initialized on ILSVRC and finetuned on RWTH with glosses.

**User studies (R3)**. Due to the diversity of translations, standard metrics may not fully reflect whether translation results convey the correct meanings or not. *To further demonstrate our performance gains*, we ask two participants to compare the predictions of TSPNet-Joint and [2] on RPWT to ground-truth translations, and choose the semantically more relevant result. The two participants favor 562 (87.54%) and 580 (90.34%) translations of TSPNet-Joint over those of [2] out of 642 testing instances. This experiment further demonstrates our significant performance improvements.

**Difference between SLT and generic action localization (R3)**. Since signing progresses continuously, there is no clear boundary between consecutive sign gestures. Thus, it is ambiguous whether each sign gesture or an entire sign sequence should be treated as an action. Moreover, natural languages differ from sign languages grammatically. As a result, segmenting continuous signing into isolated signs is very difficult, especially in the absence of glosses. In contrast, as for the action localization tasks, temporal boundaries of an activity are evidently clear. This results in the essential difference between these two tasks. Although our approach is designed for SLT, we agree with R3 that it would be interesting to see our method generalizes to weakly supervised action localization, where only action labels are provided without boundary annotations. Due to the rebuttal timeframe, we have to leave this in future work.

**Computation cost (R3)**. TSPNet-Joint and TSPNet-Single share the same architecture while TSPNet-Joint has 18% more operations for hierarchical feature learning over multi-granularity features. TSPNet-Joint takes 110 ms on average to process an RPWT video on an NVIDIA V100 GPU. We will include the discussion in the revision.

**Segmenting isolated signs (R4)**. Please note that we did not claim our method achieves accurate temporal segmentation. Due to the grammatical differences between sign languages and natural languages, it is very difficult to segment sign gestures especially when glosses are unavailable. This motivates us to explore multi-scale windowing segments to learn representative features for sign language translation in an end-to-end fashion. Furthermore, Table 3 indicates that it is non-trivial to integrate multi-scale segment features. In this regard, our hierarchical feature learning method provides an effective solution to representing gestures for sentence translation in the absence of gloss annotations.

[Meta-Review · NeurIPS 2020]

The reviewers were positive about the ideas in the paper and mostly debated the merits of the evaluation. For one they were not fully convinced about the arguments in the rebuttal about the differences between the sharpness of boundaries for action localization and sign language translation. For camera ready I would suggest better addressing this point, as well as comparing or justifying differences to "Sign Language Transformers: Joint End-to-end Sign Language Recognition and Translation", Camgoz et al, CVPR 2020. One final suggestion is to add results with one more video encoder in addition to I3D.